# Propulsive Force Modulation Drives Split-Belt Treadmill Adaptation in People with Multiple Sclerosis

**DOI:** 10.3390/s24041067

**Published:** 2024-02-06

**Authors:** Andrew C. Hagen, Christopher M. Patrick, Isaac E. Bast, Brett W. Fling

**Affiliations:** 1Department of Health and Exercise Science, Colorado State University, Fort Collins, CO 80523-1582, USA; cm.patrick@colostate.edu (C.M.P.); isaac.bast@colostate.edu (I.E.B.);; 2Molecular, Cellular and Integrative Neuroscience Program, Colorado State University, Fort Collins, CO 80523-1617, USA

**Keywords:** propulsion, split-belt treadmill, multiple sclerosis, motor adaptation, gait asymmetry

## Abstract

Most people with multiple sclerosis (PwMS) experience significant gait asymmetries between their legs during walking, leading to an increased risk of falls. Split-belt treadmill training, where the speed of each limb is controlled independently, alters each leg’s stepping pattern and can improve gait symmetry in PwMS. However, the biomechanical mechanisms of this adaptation in PwMS remain poorly understood. In this study, 32 PwMS underwent a 10 min split-belt treadmill adaptation paradigm with the more affected (MA) leg moving twice as fast as the less affected (LA) leg. The most noteworthy biomechanical adaptation observed was increased peak propulsion asymmetry between the limbs. A kinematic analysis revealed that peak dorsiflexion asymmetry and the onset of plantarflexion in the MA limb were the primary contributors to the observed increases in peak propulsion. In contrast, the joints in the LA limb underwent only immediate reactive adjustments without subsequent adaptation. These findings demonstrate that modulation during gait adaptation in PwMS occurs primarily via propulsive forces and joint motions that contribute to propulsive forces. Understanding these distinct biomechanical changes during adaptation enhances our grasp of the rehabilitative impact of split-belt treadmill training, providing insights for refining therapeutic interventions aimed at improving gait symmetry.

## 1. Introduction

Multiple sclerosis (MS) is the most common neurodegenerative disease affecting young adults, with the average onset occurring at 31 years of age [1]. As a result, most people with MS (PwMS) contend with the effects of the disease for more than half of their lifespan. MS is characterized by the degradation of the myelin sheath, a protective layer consisting of lipids and proteins that enhances the velocity of electrical impulse propagation along a nerve. Consequently, this degradation creates diminished neural communication throughout the central nervous system. Along with common symptoms of fatigue, neuropathy, and instability [2], 93.7% of PwMS report having significant mobility impairments [3] and that gait dysfunction is the leading contributor to decreased quality of life [4]. Specifically, PwMS typically have one side of the body that is more affected (MA) and one side that is less affected (LA) in strength and function. This leads to an asymmetrical gait pattern, one of the greatest risk factors for falls [5].

Successful locomotion in diverse environments requires dynamic gait pattern adaptation to ensure stability, efficiency, and safety. Perturbations during walking, such as variations in surfaces, and encounters with stairs, curbs, or entrances demand real-time adjustments in gait to navigate these challenges. Motor adaptation is a fundamental aspect of these gait adjustments, particularly when individuals encounter novel or unexpected conditions [6,7]. A common laboratory method for evaluating gait adaptability involves the use of a split-belt treadmill, where the speed of each leg is controlled independently on two separate belts [8]. When exposed to a split-belt treadmill, individuals experience a discrepancy between the movement of their two legs, creating an externally imposed perturbation. To walk successfully on the split-belt treadmill, the nervous system engages in an error correction process, continuously adjusting the timing and coordination of each limb [9]. This adaptive mechanism enables individuals to adjust their stepping and maintain balance despite the treadmill’s asymmetric speed. The adaptation process results in aftereffects, where individuals exhibit changes in their gait pattern even after returning to normal walking conditions [10]. These aftereffects suggest that the nervous system stores the adapted walking pattern using feedforward predictive modulation [11]. 

Understanding gait adaptability is essential not only for studying typical locomotion but also for developing effective rehabilitation strategies for dysfunctional gait. Previous research has demonstrated that individuals with various neurodegenerative conditions, such as stroke [12,13], Parkinson’s disease [14,15], and, more recently, PwMS [16] maintain the ability to adapt while walking on a split-belt treadmill. While many studies have explored broader assessments of gait adaptability, such as step length asymmetry [17], few have assessed the specific biomechanical changes that contribute to these broader adaptations. The aim of this study was to elucidate the specific kinematic and ground reaction force (GRF) changes that occur in PwMS during adaptation on a split-belt treadmill. Our hypothesis was that PwMS would exhibit significant changes in propulsive force generation and ankle kinematics during this adaptive process due to PwMS commonly experiencing reduced propulsion and ankle joint moments and powers during typical walking [18,19]. Investigating these specific biomechanical changes provides a more nuanced understanding of gait adaptability in PwMS and can contribute valuable insights to the development of targeted rehabilitation interventions for this population.

## 2. Materials and Methods

### 2.1. Participants

We recruited a convenience sample of participants in and around northern Colorado. Inclusion criteria required participants to have relapsing-remitting MS, be between the ages of 18 and 86, be fully ambulatory without an assistive device, and be capable of walking three-tenths of a mile (~500 m) without stopping to rest. The criteria related to MS severity required individuals to score between 2 and 5.5 on the Expanded Disabilities Status Scale (EDSS). This ensured participants had at least minimal disability but could perform the split-belt treadmill adaptation paradigm safely and with minimal fatigue. Exclusion criteria precluded participants from having a history of balance impairments or brain injury unrelated to MS, or any musculoskeletal injury or related surgical procedure which impacted gait in the previous six months. This study was approved by the Colorado State University Biomedical Institutional Review Board (protocol code 1664).

Following screening and informed consent, demographics and participant-reported disease characteristics were collected using REDCap survey software (v. 13.1.32). Surveys collected included the EDSS, Multiple Sclerosis Walking Score 12 (MSWS-12), Modified Fatigue Impact Scale (MFIS), Short Form 36 (RAND 36), Beck Depression Inventory (BDI-II), and the Montreal Cognitive Assessment (MOCA).

### 2.2. Split-Belt Treadmill Adaptation Paradigm

Participants completed five different walking trials (Figure 1). The first two overground trials consisted of two separate 2-min walk tests: one at the participant’s preferred walking speed, and the other at the fastest walking speed at which the participant was comfortable. Participants then completed a 2-min walk on the treadmill in the tied-belt configuration to set their preferred walking speed. Immediately following this, the adaptation period was initiated, and the treadmill was put into the split-belt configuration: the fast belt was set to the participants’ fast walk speed, while the slow belt was set to half the speed of the fast belt (2:1 ratio). The adaptation period concluded after 10 min, at which point the treadmill was set back to the tied-belt configuration at the participant’s preferred walking speed for a 1-min post-adaptation trial.

### 2.3. Gait Analysis

Gait cycle parameters were measured during all walking trials. Participants wore 6 APDM Opal inertial sensors (APDM Inc., Portland, OR, USA) during the overground baseline trials to determine the walking speed for subsequent trials. During the treadmill trials, participants wore 16 retroreflective markers for the collection of three-dimensional motion capture data, sampled at 100 Hz. Participants walked on a custom-built split-belt treadmill instrumented with Bertec force platforms (Model 4060-10, Bertec Corp., Columbus, OH, USA), which collected three-dimensional ground reaction force (GRF), sampled at 1000 Hz. The treadmill consists of two separate belts, each with its own motor, allowing each belt to move at independent speeds. Prior research determined that the fast belt should be under the MA limb to improve gait symmetry [20]. The MA limb was determined by participant self-reporting and investigator observation during the overground walking trials.

### 2.4. Data Processing

Three-dimensional trajectory and force data were processed using Vicon Nexus software (v2.14, Vicon Motion Systems, Oxford, UK). Trajectory positions were filtered using a fifth-order spline-interpolating Woltring filter [21] implemented in Vicon Nexus and utilizing a generalized cross-validation approach, and joint kinematics were calculated using the Vicon Plug-In Gait modeling pipeline. Gait cycle events were identified using a custom MATLAB script (v. 9.13.0, MathWorks Inc., Natick, MA, USA) implemented in Vicon Nexus with a resultant force threshold of 25 N. Kinematic data were divided into stance and swing phases and interpolated to 100 values per phase. Force data were filtered using a fourth-order zero-lag Butterworth filter with a low-pass cutoff frequency of 300 Hz. Force data were also interpolated to 100 values, but only for the stance phase. 

Kinematic and GRF data were averaged within each limb for select gait cycles during the treadmill walking trials. A total of five timepoints were defined as follows: Baseline (last 15 gait cycles of 2-min tied belt trial), Early Adapt (gait cycles 6–15 of split-belt adaptation trial), Late Adapt (last 15 gait cycles of split-belt adaptation trial), Early Post-Adapt (gait cycles 6–15 of post-adaptation trial), and Late Post-Adapt (last 15 gait cycles of post-adaptation trial). The choice of gait cycles 6–15 was made to ensure the treadmill belts had reached the appropriate speed and steps were consistent [22]. The choice of the last 15 gait cycles was to capture the gait cycles in which participants were most practiced in the respective configuration (i.e., tied- and split-belt) and to normalize them for different walking speeds across participants.

Primary outcome variables included joint range of motion and joint angles for the hip, knee, and ankle, as well as braking, propulsion, and vertical GRF, normalized to a percentage of participant body weight (%BW) of the MA and LA limbs. Using the trapezoidal method of integration, impulse (%BW × %Stance) was calculated for the propulsive portion of the stance phase [23]. From these, asymmetry values were calculated as MA − LA at each timepoint, with a positive value indicating a larger value for the MA limb. For more information on step length asymmetry and other more global measures of adaptation for this sample, we refer the reader to a previous publication [16].

### 2.5. Statistical Methods

Following processing, data were imported into R Statistical Software (v4.2.1; R Core Team, Vienna, Austria, 2022) to complete statistical analysis. A 2 × 5 repeated measures analysis of variance (RMANOVA) was calculated for each outcome of interest with limb (MA vs. LA) as a between-factor and timepoint as a within-factor. Residuals versus fitted plots along with quantile–quantile plots were used to confirm normality and equal variance, while Mauchly’s test was used to confirm sphericity. Following the observation of a significant main effect and an interaction between limb and timepoint, post hoc pairwise comparisons were conducted using false discovery rate (FDR) to correct for multiple comparisons. The reported *p*-values (*p*) are the results from each pairwise comparison with FDR corrections applied. Effect sizes (d) were calculated using Cohen’s d and correlations were calculated using Pearson’s product-moment correlation coefficient (r).

## 3. Results

### 3.1. Participant Characteristics

In total, 32 participants (69% female) with MS completed this study. The mean participant age was 50.4 (12.0) years with an average of 12.7 (8.6) years since diagnosis. A total of 84% of participants reported neuropathy. A mean MSWS-12 of 22 (12) and a mean EDSS of 3.7 (0.8) indicated that MS symptoms among this cohort were mild [24]. Further, participants were more active than normative PwMS [25], exercising 329 min per week on average. Table 1 provides a comprehensive description of participant characteristics.

### 3.2. Peak Propulsion and Dorsiflexion Asymmetry Adaptation

Throughout the time course of the split-belt treadmill adaptation paradigm, participants experienced the largest changes in peak propulsion asymmetry and peak dorsiflexion asymmetry. For peak propulsion, the MA limb immediately produced more propulsion at Early Adapt compared to Baseline (*p* < 0.001, d = 1.40). Subsequently, PwMS progressively enhanced propulsion production from Early Adapt to Late Adapt (*p* < 0.001, d = 0.56). Following this, the LA limb produced a more propulsive force than the MA limb (*p* < 0.001, d = 0.79), demonstrating a negative aftereffect which is indicative of successful adaptation (Figure 2).

Further results suggest that a reduction in dorsiflexion is a primary contributor to the heightened propulsion observed during adaptation. This inference is supported by a robust negative correlation between propulsion asymmetry adaptation and dorsiflexion asymmetry adaptation (r = −0.86, *p* < 0.001). For peak dorsiflexion, the MA limb had an immediate decrease at Early Adapt compared to Baseline (*p* < 0.001, d = 0.97). Subsequently, from Early Adapt to Late Adapt, the MA limb exhibited a progressive decline in dorsiflexion (*p* = 0.0031, d = 0.38). Following this, the MA limb produced more dorsiflexion than the LA limb during Early Post-Adapt (*p =* 0.0051, d = 0.55), demonstrating a negative aftereffect and confirming the occurrence of adaptation (Figure 2).

### 3.3. Peak Propulsion Adaptation

To assess force profile changes across the experimental paradigm, peak propulsive force, peak braking force, peak early vertical GRF, and peak late vertical GRF were collected and analyzed for each limb (Figure 3, Appendix A). For the LA limb, peak propulsion force is reduced during Early Adapt compared to Baseline. Subsequently, peak propulsive forces increase in Late Adapt compared to Early Adapt. However, the shape of the data curve during the adaptation period is markedly flat. Considering the flat curve in conjunction with the absence of aftereffects indicates that there is insufficient evidence to conclude that propulsive adaptation is occurring in the LA limb. Peak braking, peak early vertical GRF, and peak late vertical GRF in the LA all follow a similar pattern where there are no changes during the adaptation period, indicating that changes from baseline are reactive rather than adaptive. 

For the MA limb, all four force profiles display a reduction in force during Early Adapt compared to Baseline (Propulsion: *p* = 0.0011, d = 0.39|Braking: *p* < 0.001, d = 1.16|Early Vertical GRF: *p* < 0.001, d = 0.81 | Late Vertical GRF: *p* < 0.001, d = 1.22) and a subsequent steady increase in force production during the adaptation period, resulting in increased force during Late Adapt compared to Early Adapt (Propulsion: *p* < 0.001, d = 0.81 | Braking: *p* < 0.001, d = 1.51 | Early Vertical GRF: *p* = 0.0027, d = 0.45 | Late Vertical GRF: *p* < 0.001, d = 0.72). Notably, during the Post-Adapt period, peak propulsion presents a negative aftereffect where propulsion during Early Post-Adapt is reduced compared to propulsion during Baseline (*p* < 0.001, d = 0.48), and, further, propulsion during Late Post-Adapt is increased relative to Early Post-Adapt and approaching Baseline. Together, the peak propulsion curve during the adaptation period and evidence of negative aftereffects highlight peak propulsion in the MA limb as a key contributor to gait adaptation. Peak braking, peak early vertical GRF, and peak late vertical GRF in the MA limb fail to exhibit the same aftereffects, and thus despite having promising adaptation curve profiles, they likely contribute to adaptation but are not the main driving characteristics underlying gait adaptation in PwMS.

### 3.4. Peak Joint Angle Adaptation

Examining each limb individually reveals insights into peak joint angles across gait cycles, revealing distinct trends for the MA and LA limbs (Figure 4, Appendix A). For the LA limb, nearly all joint motions exhibited a reduction in peak angles from Baseline to Early Adapt. Subsequently, during the adaptation period, no further changes occurred, leading to the absence of aftereffects during Early Post-Adapt compared to Baseline (Appendix A). This finding indicates that the changes experienced in the LA limb are reactive rather than adaptive. LA peak dorsiflexion was the only LA joint angle that increased, but it followed a similar reactive feedback pattern to all other joints. In contrast, the MA limb exhibited notable adaptive changes, particularly in peak ankle plantarflexion. Peak plantarflexion demonstrated an immediate increase from Baseline to Early Adapt (*p* = 0.038, d = 0.23), followed by a slight decrease from Early Adapt to Late Adapt (*p* = 0.013, d = 0.22), while still maintaining heightened plantarflexion relative to Baseline. A negative aftereffect in peak plantarflexion further confirmed the occurrence of adaptation (*p* = 0.0019, d = 0.35). There was an immediate decrease in MA peak dorsiflexion from Baseline to Early Adapt (*p* = 0.0012, d = 0.38), followed by a continued reduction in dorsiflexion from Early Adapt to Late Adapt (*p* = 0.078, d = 0.24) that failed to reach significance, likely due to high individual variability at Late Adapt. Further, the negative aftereffect also failed to reach significance during Early Post-Adapt (*p* = 0.068, d = 0.26). Additionally, peak hip extension increased during adaptation (*p* < 0.001, d = 0.44), but there was no significant aftereffect during Early Post-Adapt. An examination of joint ROM across the split-belt treadmill time series revealed minimal changes in the MA limb and decreases in all LA joint ROM during Early Adapt and Late Adapt, without an aftereffect during Early Post-Adapt (Appendix A). This suggests that the reduction in joint ROM for the LA limb is reactive rather than adaptive and likely attributable to a slower belt speed under the LA limb.

### 3.5. Propulsion Impulse and Ankle Joint Profile Adaptation

With propulsion in the MA limb identified as a key contributor to gait adaptation, propulsion impulse in the MA limb was analyzed as force (% BW) over time (% Stance) to obtain a more detailed assessment of propulsion adaptation over the split-belt treadmill paradigm. Despite peak propulsion being less at Early Adapt compared to Baseline in the MA limb, there is no significant change in propulsion impulse from Baseline to Early Adapt (*p =* 0.54, d = 0.075). There is subsequently more propulsion impulse at Late Adapt compared to Early Adapt, showing a steeper rate of force development at Late Adapt (*p* < 0.001, d = 0.47). Early Post-Adapt has reduced propulsion impulse compared to Baseline (*p* < 0.001, d = 0.49) and furthermore displays similar negative aftereffects to peak propulsion. 

An examination of joint profiles throughout the split-belt adaptation paradigm revealed significant changes in peak dorsiflexion timing within the gait cycle. Specifically, peak dorsiflexion in the MA limb occurred 22.3% earlier in the gait cycle during Early Adapt (*p* < 0.001, d = 1.64) when compared to baseline and 17.1% earlier during Late Adapt (*p* < 0.001, d = 1.36) when compared to baseline, signifying an accelerated onset of plantarflexion (Figure 5). This temporal shift, along with related internal musculoskeletal changes, allowed for an extended period of plantarflexion during the gait cycle, facilitating enhanced propulsive force production. Additional joint profiles demonstrated that the LA limb had a decreased range of motion during adaptation (Appendix A).

## 4. Discussion

In this study, PwMS completed a split-belt treadmill adaptation paradigm, and the biomechanical changes that occurred were investigated. The primary finding was that peak propulsion asymmetry between the limbs increased from Early Adapt to Late Adapt and presented a substantial aftereffect from Baseline to Early Post-Adapt. Additionally, the largest kinematic contributors to these increases in peak propulsion asymmetry were peak dorsiflexion asymmetry, increased MA propulsion impulse, and earlier onset of MA plantarflexion in the gait cycle at the same timepoints. Further, both the kinematic and GRF changes in the MA limb (under the fast belt) experienced predictive feedforward adaptation, while the changes in the LA limb (under the slow belt) only experienced reactive feedback adjustments.

### 4.1. Increased Propulsive Forces Drive Split-Belt Treadmill Adaptation in PwMS

Previous literature has established that PwMS exhibit decreased propulsion and decreased ankle power during gait, even in mildly impaired individuals [18,19,26]. Both negative and positive ankle plantarflexion power have been shown to be lower in PwMS compared to healthy controls [26]. Further, positive ankle plantarflexion power has been identified as a robust predictor of step length and walking speed and likely contributes to a reduction in propulsion in PwMS [18,27]. In response to these findings, research efforts have focused on ankle rehabilitation in PwMS to facilitate toe push-off, leading to increased propulsive force during gait and a faster walking speed [28]. Similar work has aimed to augment propulsion by increasing propulsive demand by walking at an incline and walking with backward-directed resistance in clinical populations and healthy controls [29,30,31]. With the establishment that neuromuscular function surrounding the ankle is key for propulsion and reduced propulsion impairs gait in PwMS, our study looked to characterize propulsion adaptation during split-belt treadmill training. 

Specifically, peak propulsion asymmetry values initially indicate that the LA limb produced more anterior force at Baseline. At the onset of the adaptation period, as the belt speeds changed, they increased the propulsive demand for the MA limb and decreased the propulsive demand for the LA limb. As a result, peak propulsion asymmetry values crossed zero, indicating that the MA limb produced more propulsive force, which was expected. The MA limb continued to increase propulsive force for the duration of the adaptation period while the LA limb remained relatively unchanged, creating a robust adaptation curve. Finally, during Post-Adapt, asymmetry values displayed aftereffects, indicating successful adaptation. Thus, increasing the speed of the belt under the MA limb increased the propulsive demand for the MA limb leading to the adaptation of propulsive force generated by the MA limb. These results add to a growing body of literature identifying propulsion as a key factor of gait rehabilitation in clinical populations.

### 4.2. Decreased Dorsiflexion and Early-Onset Plantarflexion Modulate Propulsion

It is likely that the observed enhancement in propulsion during adaptation is attributed to a unique strategy of joint motion changes tailored to each participant. Most notably, the musculoskeletal changes that resulted in altered peak dorsiflexion asymmetry and the timing of plantarflexion onset had the largest impact overall. Peak dorsiflexion asymmetry between the limbs had a robust correlation across the split-belt treadmill adaptation time course with peak propulsion asymmetry (r = −0.86, *p* < 0.001), implying the importance of reducing peak dorsiflexion as a key driver for heightened propulsion. Specifically, previous literature has identified that excessive dorsiflexion during stance is common in PwMS and hinders propulsion; thus, reducing peak dorsiflexion may be a direct mechanism to facilitate propulsion [18]. Consistent with this observation, plantarflexion onset occurred earlier during the stance phase in the adaptation period, allowing for a longer period of plantarflexion during the gait cycle, and, as a result, generating heightened peak propulsion at toe off. This is further corroborated by observing the propulsion impulse profile, which exhibits a steeper and quicker development or propulsive force during the adaptation period (Figure 5). 

This finding of an earlier plantarflexion onset is contrary to other split-belt treadmill research in young, healthy individuals, suggesting that PwMS have an altered adaptive strategy [32]. Interestingly, one study indicated that peak dorsiflexion time (and therefore plantarflexion onset) occurred sooner only at a 4:1 speed ratio and not at lower speed ratios [22]. Given that this study employed a speed ratio of 2:1, it implies that PwMS may adopt this strategy even at a milder belt speed ratio. This adaptation strategy also may be driven by compromised propulsive forces during normal walking in PwMS [18], necessitating an increased propulsion demand for successful adaptation. Additionally, this observation raises the possibility that the demands faced by PwMS using a 2:1 belt speed ratio are more akin to those encountered by healthy individuals using a 4:1 belt speed ratio.

Following an immediate increase at Early Adapt, peak plantarflexion also adapted and was reduced from Early Adapt to Late Adapt. While this may seem contradictory, due to plantarflexion occurring earlier in the gait cycle, continual peak plantarflexion increases are not necessary to continue to produce more propulsion. Moreover, this may suggest that soft tissue loading, by primarily the Achilles tendon, may be an additional strategy to generate more propulsive force [33]. Further, PwMS experience heightened coactivation of the gastrocnemius and tibialis anterior as a result of spasticity and the prioritization of ankle joint stability [34]. This may serve as a plausible explanation for the concurrent reduction in both plantarflexion and dorsiflexion observed during adaptation.

Secondarily, hip extension may assist with propulsion increases during adaptation, but not adapt by itself. In accordance with these data, it can be inferred that split-belt treadmill adaptation likely occurs distal to proximal due to the perturbation being applied to the feet, thereby creating the largest effect at the ankle joint and only residual effects at the hip joint (Appendix A).

### 4.3. The MA Limb Engages in Predictive Feedforward Mechanisms while the LA Limb Relies on Reactive Feedback Mechanisms

Split-belt treadmill adaptation is contingent on a combination of reactive feedback control and predictive feedforward modulation. Findings from lesion studies [35] indicate that immediate gait parameter adjustments utilize feedback mechanisms independent of supraspinal control and are likely mediated through the brainstem and central pattern generators. However, to sustain the effects of split-belt treadmill adaptation and generate feedforward aftereffects, supraspinal influence from the cerebellum [36] and the cerebrum [37] is imperative. Furthermore, it has been recently suggested the microstructural properties of the inferior cerebellar peduncle are associated with locomotor adaptability [38]. In PwMS, the inferior cerebellar peduncle has shown compromised microstructural integrity [39], which may explain the increased demands observed in PwMS at a 2:1 belt ratio that are absent in neurotypicals until a 4:1 belt speed ratio.

The most prominent adaptive changes observed during the split-belt adaptation paradigm were in peak propulsion asymmetry and peak dorsiflexion asymmetry. The existing literature strongly supports the notion that interlimb parameters are the primary components that undergo adaptation, while intralimb parameters primarily undergo adjustments through reactive feedback but do not exhibit adaptation or aftereffects [10]. The present study aligns with this, as the largest effect sizes of adaptation and aftereffects were identified in propulsion asymmetry and peak dorsiflexion asymmetry between the limbs. This is consistent with the prevailing understanding of interlimb adaptability.

Notably, when examining individual joint-level changes in each limb, the data suggest that parameters in the LA limb (slow belt) predominantly rely on reactive feedback, while parameters in the MA limb (fast belt) undergo adaptation. This suggests that the MA limb plays a primary role in driving the significant adaptive changes observed in peak propulsion asymmetry and peak dorsiflexion asymmetry. It has been historically understood that intralimb parameters, including MA peak propulsion and MA peak plantarflexion, do not adapt during split-belt treadmill adaptation. However, other research in healthy controls has proposed that unlike global intralimb parameters, such as stride length, that are dependent on multiple coordinated joint motions, individual intralimb joints are able to adapt [22], and this finding is corroborated by the presented data.

### 4.4. Limitations

A primary limitation of this study, and many studies involving PwMS, is the considerable distribution of symptom severity. Evidence has indicated that cutaneous perception in the feet plays a role in adaptation, particularly regarding force perception [32]. Many PwMS report neuropathy in their distal limbs [40], which could impact the biomechanical strategy used to adapt on a split-belt treadmill. Additionally, age is known to affect gait adaptability during split-belt treadmill adaptation [41]. Given the wide age range of PwMS in this study, the effects of age alone could potentially impact adaptability, independent of MS-related factors. Further, this sample of PwMS was quite active and relatively healthy compared to normative PwMS. Most participants had a lower disability level than average compared to other PwMS with a similar number of years since diagnosis. However, a fundamental constraint of split-belt treadmill adaptation is the requisite capability to complete the demanding walking task, which unfortunately excludes many individuals with a higher disability level and limits the generalizability of the findings. With this noted, it is possible that as technology and methodologies improve over time, allowing for the study of gait adaptation in individuals with more severe disability, larger effect sizes may be observed from a study with a similar methodology. An additional improvement to this study could involve incorporating a longer tied-belt walking session following split-belt treadmill adaptation. This would enable the recording of the rate of deadaptation after the observed aftereffects and offer valuable insights into the biomechanical strategies and savings during deadaptation.

### 4.5. Future Work

This study presents evidence indicating that propulsive force amplification, specifically generated by the MA limb, has the capacity to augment gait adaptation during split-belt treadmill training in PwMS. Therefore, future work should expand on these findings by implementing interventions directly aimed at training and bolstering propulsive force in the MA limb in PwMS combined with, or prior to, split-belt treadmill training. Previous literature has identified several ways to manipulate propulsion during gait training in clinical and neurotypical populations, including inclined walking [29,30], walking with backward-directed resistance [31], and functional electrical stimulation [42,43]. We anticipate that findings from future studies, paired with the present results, will further highlight the importance of propulsive force in gait adaptation in PwMS and promote the need for propulsive intervention in rehabilitation strategies.

To further build on this work, future research should investigate the neural underpinnings that generate split-belt treadmill adaptation, specifically within PwMS. Previous investigations have highlighted the cerebellum [36,44,45], along with the cortico-cerebellar and cortico-striatal loops [46] as principal modulators of motor adaptation. Notably, research has indicated that PwMS exhibit impaired cerebellar communication [39,47], adding complexity to their adaptive processes. Given the observed similarity in demands for PwMS at a 2:1 belt ratio compared to neurotypical individuals at a 4:1 belt speed ratio [22], the identification of the neural substrate responsible for these altered demands becomes a pertinent avenue for exploration. Moreover, determining the neural underpinnings of these demand changes offers valuable insights into optimizing the clinical efficacy of motor adaptation in PwMS, thereby informing targeted interventions and improving rehabilitation outcomes.

## 5. Conclusions

In conclusion, this study investigated the biomechanical adaptations that occurred during split-belt treadmill walking in PwMS. The primary finding was a significant increase in peak propulsion asymmetry between the limbs, with the main contributors being peak dorsiflexion asymmetry and an earlier onset of plantarflexion in the gait cycle. Additionally, this study highlighted that the MA limb played a predominant role in driving adaptive changes, relying on predictive feedforward mechanisms, while the LA limb primarily underwent reactive feedback adjustments. Moreover, this study underscores the importance of gaining further insights into the biomechanics of split-belt treadmill adaptation to fully comprehend its rehabilitative potential. A more comprehensive understanding of the intricacies of gait adaptation in PwMS and other populations may contribute to the development of effective and targeted rehabilitation interventions, thereby improving mobility and quality of life for the millions of individuals living with gait dysfunction.

## Figures and Tables

**Figure 1 sensors-24-01067-f001:**
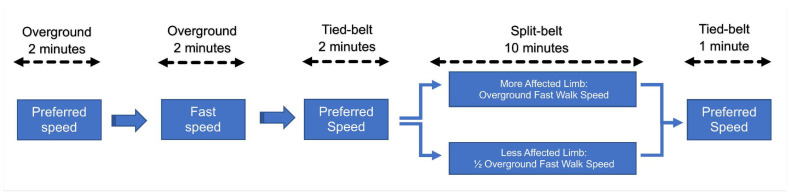
Participants completed 5 independent walking trials including baseline overground walking, fast overground walking, baseline tied-belt treadmill walking, split-belt treadmill walking, and post-adaptation tied-belt treadmill walking.

**Figure 2 sensors-24-01067-f002:**
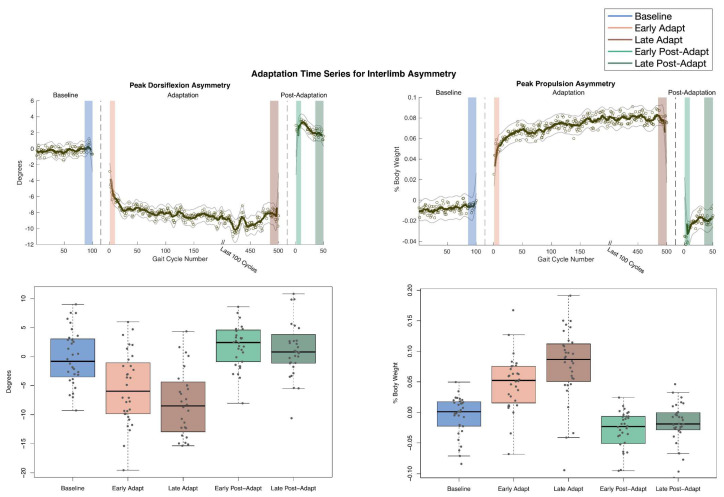
Peak propulsion and peak dorsiflexion asymmetry for each gait cycle throughout the split-belt treadmill adaptation time course averaged across all participants. Peak propulsion increased while peak dorsiflexion decreased during adaptation and generated aftereffects. Below the adaptation curves are boxplots containing each participant’s data for the specified timepoint.

**Figure 3 sensors-24-01067-f003:**
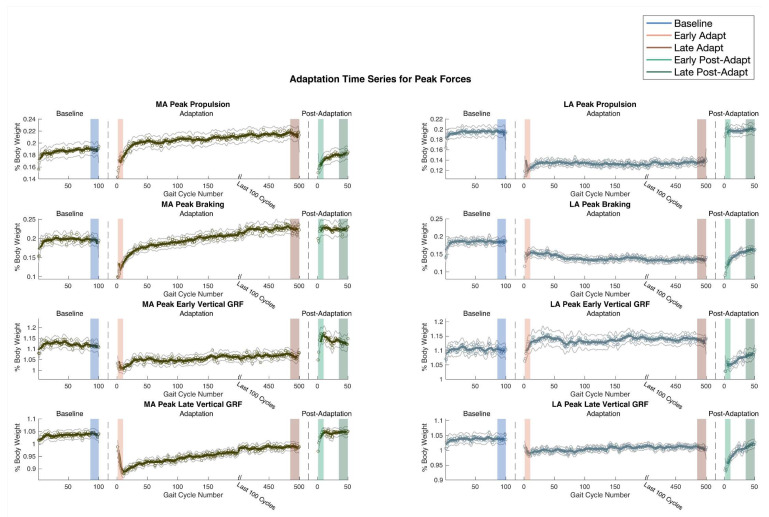
Peak propulsive, braking, early vertical GRF, and late vertical GRF, averaged across all participants, for the MA and LA limb throughout Baseline, Adaptation, and Post-Adaptation. Light grey lines indicate standard error for each point across all participants.

**Figure 4 sensors-24-01067-f004:**
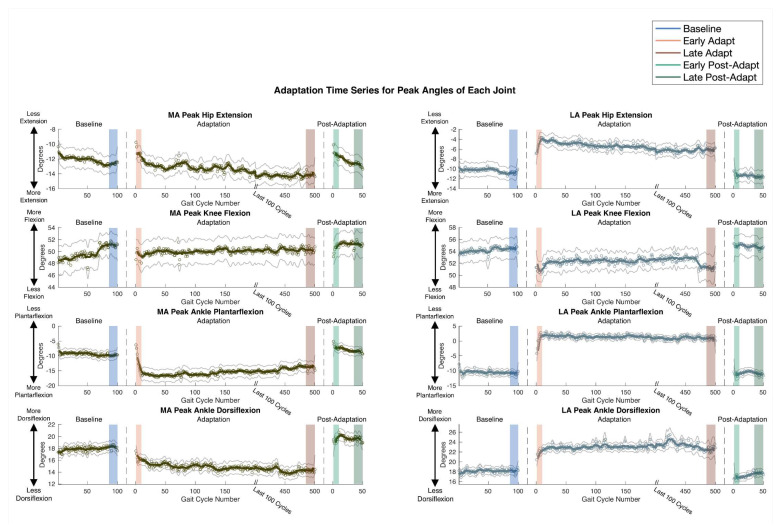
Peak joint angles, averaged across all participants, for the MA and LA hip, knee, and ankle throughout Baseline, Adaptation, and Post-Adaptation. Light grey lines indicate standard error for each point across all participants. MA peak plantarflexion experienced significant adaptative changes while all LA joints experienced significant reactive changes.

**Figure 5 sensors-24-01067-f005:**
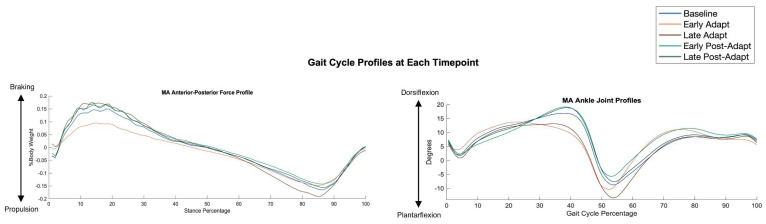
MA anteroposterior (AP) and MA ankle joint profiles, averaged across all participants for each timepoint during split-belt treadmill adaptation. At Early Adapt, the ankle joint reaches its peak dorsiflexion angle sooner and thus initiates plantarflexion earlier in the gait cycle. At Late Adapt, while the time of plantarflexion onset is adapting back toward baseline, the increase in plantarflexion is maintained.

**Table 1 sensors-24-01067-t001:** Participant characteristics. Mean and SD or proportion of cohort are reported for select attributes. Collectively, this cohort of PwMS was highly active and reported mild symptoms.

Attribute	Mean	SD
N	32	
Age	50.4	12.0
Sex	69% Female	
BMI	24.8	3.6
Activity (min per week)	329	231
Years Since Diagnosis	12.7	8.6
Number of Falls in Prior 6 Months	0.3	0.8
Reported Neuropathy	84%	
EDSS	3.7	0.8
MFIS	33	15
RAND 36: Physical Function	78	22
MSWS-12	22	12
BDI	8	7
MOCA	27	2

## Data Availability

The data presented in this study are available on request from the corresponding author. The data are not publicly available to protect participant privacy.

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
