# Peer review of "Propulsive Force Modulation Drives Split-Belt Treadmill Adaptation in People with Multiple Sclerosis"

_sensors, 2024, doi:10.3390/s24041067_

Round 1

Reviewer 1 Report

Comments and Suggestions for Authors

This paper examines kinematics and kinetics of persons with MS walking on the split belt treadmill. The authors find that adaptation occurs primarily in the fast leg = affected limb and includes changes in propulsion, peak dorsiflexion, and timing of plantatflexion initiation. Overall this is a well-presented paper that shows new data and I suggest minor revision. Specific comments follow:

My biggest concern is not under the authors control, but the plots should be larger. I assume there will be high quality versions available for download but in the review copy some text was barely legible even when zoomed in. 

There is an individual who has a significantly different strategy for the split belt, who is producing substantially more propulsion on the slow leg at peak. I wonder if the authors looked at their kinematics and kinetics in more detail to see how/why they differ. 

175: Two notes. First, I think some version of the description from 266-270 should be bumped up to here to clearly state the authors argument about how dorsiflexion and propulsion are related. Second, this may be more of a style note, but here and elsewhere in the manuscript (322) the authors describe gait in a way that implies that joint kinematics drive kinetic changes. I know the authors understand that musckuloskeletal changes, especially muscle recruitment and timing, are coordinated to produce internal kinetic changes which have joint motion and external kinetic outputs. I understand the authors aren't measuring anything directly about muscles which may be why the results are discussed in this way, but I found it jarring to read. I would reword just to make clear that internal changes drive resultant kinematic and kinetic changes, which of course do relate to each other.

Figure 5 appears to be being cropped in error. Additionally there are boxes to help the reader interpret the Y axes, but they have a white background that sometimes occludes axes labels.  

292: I don't believe the A1 and A2 abbreviations are used more than once. This does not seem to warrant abbreviation. 

Reviewer 2 Report

Comments and Suggestions for Authors

The manuscript “Propulsive Force Modulation Drives Split-belt Treadmill Adaptation in People with Multiple Sclerosis” reports a study of mechanisms which are responsible for the improvement of gait symmetry during split-belt treadmill (a treadmill which sets up the speed of each leg independently).

Asymmetrical gait pattern is common for people with multiple sclerosis and is one of the greatest risk factors for falls. The split-belt treadmill adaptation results in positive aftereffects, and individuals exhibit changes in their gait pattern. Understanding gait adaptability is essential not only for studying typical locomotion but also for developing effective rehabilitation strategies for dysfunctional gait.

The results are interesting, but there is no development of sensors. As I understood, commercially available sensors & software systems were used to capture the data. Therefore, the authors should give a reason why the paper should be published in Sensors.

I have also a few requests for authors if the manuscript would be further considered for publication:

- Please give the distance “three-tenths of a mile” in meters

- Please give details or reference concerning the Woltring filter

- In Section 3.1 and Table 1 the numbers contain too many decimal places. Please remove unnecessary digits which are smaller than statistical errors (practically all decimal places should be removed).

- Figure 2: upper part has very low quality, while lower part is in vector format and has perfect quality. Please make all figures in vector format (Figures 3-4 and Figures S1-S2 are also of poor quality). Data on the Figure 5 is cropped, please correct.

- Table S1 is cropped, please correct.

Reviewer 3 Report

Comments and Suggestions for Authors

I have had the pleasure of reviewing your manuscript titled "Propulsive Force Modulation Drives Split-belt Treadmill Adaptation in People with Multiple Sclerosis". It is a well-conducted and insightful study that significantly contributes to our understanding of gait adaptability in people with multiple sclerosis (PwMS) during split-belt treadmill walking. Below, I provide a few minor suggestions for revisions, which I believe will further enhance the clarity and impact of your manuscript.

  • While your methodology is robust, providing more comprehensive details would enhance replicability. It would be beneficial to include more information on participant selection criteria, particularly regarding the severity of MS and any comorbid conditions.
  • The current participant pool, seems to represent a relatively healthy and active subset of PwMS. Considering MS's varied impact, including participants with a broader range of disability levels could provide more generalizable results. Discussing how your findings might translate to the broader MS population would be valuable, if possible.
  • The manuscript would benefit from a more detailed discussion of how these findings can be practically applied in clinical settings. Specific recommendations or proposed modifications to current rehabilitation strategies for PwMS would be a valuable addition.
  • The discussion provides a good connection between your results and existing literature. However, it would be strengthened by addressing potential contradictions or deviations from previous studies and hypothesizing reasons for these differences.
  • Suggestions for future research perhaps including potential study designs or specific research questions, would be valuable for guiding future work in this area.
  • In my version of the paper the figure resolution is low and not well readable. Please, consider improving the quality of figures and tables.
  1.  
